# Temperature Sensing Shape Morphing Antenna (ShMoA)

**DOI:** 10.3390/mi13101673

**Published:** 2022-10-04

**Authors:** Wenxin Zeng, Wei Wang, Sameer Sonkusale

**Affiliations:** 1Department of Electrical and Computer Engineering, Tufts University, Medford, MA 02155, USA; 2Nano Lab, Tufts University, Medford, MA 02155, USA

**Keywords:** antenna, shape memory alloy, SMA, shape morphing antenna

## Abstract

Devices that can morph their functions on demand provide a rich yet unexplored paradigm for the next generation of electronic devices and sensors. For example, an antenna that can morph its shape can be used to adapt communication to different wireless standards or improve wireless signal reception. We utilize temperature-sensitive shape memory alloys (SMA) to realize a shape morphing antenna (ShMoA). In the designed architecture, multiple conjoined shape memory alloy sections form the antenna. The shape morphing of this antenna is achieved through temperature control. Different temperature threshold levels are used for programming the shape. Besides its conventional use for RF applications, ShMoA can serve as a multi-level temperature sensor, analogous to thermoreceptors in an insect antenna. ShMoA essentially combines the function of temperature sensing, embedded computing for detection of threshold crossings, and radio frequency readout, all in the single construct of a shape-morphing antenna (ShMoA) without the need for any battery or peripheral electronics. The ShMoA can be employed as bio-inspired wireless temperature sensing antennae on mobile robotic flies, insects, drones and other robots. It can also be deployed as programmable antennas for multi-standard wireless communication.

## 1. Introduction

Conventional electronic devices are structurally fixed constructs. They are realized using fixed discrete or microelectronic circuit components, each performing a unique function. For example, an integrated sensor platform consists of a dedicated sub-unit for sensing, amplification, signal conditioning, computing, memory, and RF communication. The shape and dimensions of these electronic devices are not user-controllable parameters. However, we believe this is a missed opportunity. The 3D shape of an electronic/RF component offers an often-ignored dimension for programmability and tunability that can endow new and improved functions not possible before and may even allow one to integrate multiple electronic functions in a single device. More specifically, for an integrated sensor system, a single circuit element that can perform the various functions of sensing, computing, memorizing, and communication would be a significant advance, with an advantage in size, weight, area, and power. In this paper, we discuss such a possibility of multi-functionality endowed by a ShMoA.

Being an essential component in a wireless communication system, antennas have various forms and attributes that are designed for a specific target metric. For example, an antenna may be designed for beamforming or omnidirectional transmission. It may be designed for a particular frequency band of the wireless spectrum. While there are few examples of a multifunctional antenna used in a manner discussed in this paper, there are, however, several examples in the literation for reconfigurable antennas. Printed planar antennas with discrete sections connected by diodes are one of the most common ways to achieve reconfigurability [1,2,3,4]. By turning on and off diode switches, the effective length, and geometry of the antenna change, resulting in different resonating frequencies. The same method has been utilized to configure the antenna between broadband and single band [5,6] and radiation pattern [7]. Diode switching is simple in design and fabrication, but DC biasing network to control operation adds another complexity. Another popular approach to realizing reconfigurable antennas is to utilize origami, where conductive materials are attached to a flexible substrate. The antenna’s geometry is changed through the folding and unfolding of the substrate [8,9,10]. Cubical and helical 3D structures have also been proposed with specific folding techniques to reconfigure radiation patterns and frequency [11,12,13,14,15]. Although easy to configure, origami antennas cannot alter shape independently and need external force such as actuators. Liquid metal or conductive liquid have been studied as another switching component. With the advantage of fluidity, sections of the antenna can be connected or disconnected, with the liquid metal flowing in and out of place [16,17,18,19,20]. Antennas entirely composed of liquid metal inside micro-channels are proposed as well [21,22,23]. Other mechanics utilizing MEMS structure [24,25] or the physical changes depending on applications have been presented. Such RFID antennas can alter state depending on orientation, vibration, or an approaching magnet [26,27,28,29,30,31,32]. Herein, we propose an alternative approach to realize a multi-functional reconfigurable antenna that integrates both shape morphing and actuation in the structure of the antenna using shape memory alloys. In addition, we show for the first time that ShMoA can perform embedded sensing, memory, and computing through their temperature-dependent shape change.

Shape memory alloy (SMA) are key building block for the realization of ShMoA. SMAs are typically metal alloys such as nickel–titanium that change their shape according to the applied temperature and history. At a certain temperature, the SMA can return to its original shape, which is memorized within the material and can be thermo-mechanically trained [33]. This thermo-mechanical memory of the SMA was used for the realization of a reconfigurable antenna in related work [34]. However, it did not demonstrate any additional functionality such as sensing or memory. In our prior work, we demonstrated temperature sensing using SMA-based antennas [33,35,36,37,38,39] with integrated memory of temperature events in the shape of the SMA. In these works, the SMA antenna irreversibly changes its configuration from parallel to dipole when the temperature rises above the threshold value altering its radiation profile. Interestingly, the antenna stays locked in this configuration even after the temperature drops back below the threshold temperature. In essence, this sensor memorizes the temperature crossing event without an electronic memory device or a power supply. However, its usage is limited to the detection of a single temperature threshold crossing. In this article, we leverage the antenna’s temperature-dependent shape morphing capability to perform multi-level temperature sensing for the first time; multi-level sensing essentially combines sensing with logical computing and memory, all embedded within an SMA-based antenna structure, with a simple readout through RF interrogation. Similar to an insect’s antenna used for thermal perception, the SMA based ShMoA can also be employed as temperature sensing antenna on robotic flies, insects, drones and other robots for search and rescue operation. See Figure 1a for a suggested application of ShMoA as an insect antenna.

## 2. Principle of ShMoA

### 2.1. State Changing of SMA

SMA are metals that have a memory effect depending on environmental temperature. Most SMAs are alloys made of nickel and titanium. At low temperatures, the SMA is in its martensite state. Atoms are loosely patterned in this state so that the shape of SMA can be easily modified. While at high temperature, the SMA is in the austenite state, where the atoms are tightly arranged, making the whole SMA stiff and rigid. When the temperature crosses the threshold level, the heat absorbed by the SMA is sufficient for its crystal structure to rearrange, generating strain within the material and allowing it to recover to its preprogrammed shape, regardless of its previous deformed shape, which represents the inherent memory of the SMA [40,41]. Due to the hysteresis of the SMA, the threshold temperatures of state-change are different for increasing and decreasing temperature [42,43]. As shown in Figure 1b, the start and end temperatures for the state-changing from martensite to austenite are A_s_ and A_f_, respectively; the start and end temperatures for the state changing from austenite to martensite are M_s_ and M_f_, respectively [44]. Figure 1c shows the photo of SMA wires in austenite and martensite states.

This controlled shape change of SMA can be used to make ShMoA that change their shape and morphology as a function of temperature. To alter shape according to different levels of temperature events, two sections of SMA wire with different temperature thresholds conjoined to form the SMA-based antenna. Each section will recover to its predefined configuration when the temperature exceeds its threshold, respectively, which leads to different radiations. By sensing and differentiating the radiation information, the critical historical temperature information is memorized (recorded), which can be read out at any later date using a radiofrequency readout. Beyond sensing, the temperature can also be used as a stimulus to tune/adapt the antenna behavior (frequency response, impedance matching, etc.) in wireless communication applications.

### 2.2. Working Mechanism of ShMoA

The design of the ShMoA tunes the return loss of an antenna based on its configuration; it consists of two SMA wire arms, as shown in Figure 2. Each arm comprises one red color section made of an SMA exhibiting a low-temperature threshold at T_1_ and one blue color section fabricated with an SMA with a high-temperature threshold of T_2_. When the temperature is less than T_1_, the whole sensor is manually reset to the shape shown in Figure 2a. When the temperature rises above T_1_, the red section recovers its predefined shape as a straight line, which results in a dipole antenna configuration, as shown in Figure 2b. The shape reconfiguration alters the radiation profile of the antenna and, consequently, its reflected S-parameter S_11_. Even if the temperature drops below T_1_, the antenna still holds the dipole antenna configuration. This can be used to ensure antenna design is robust to temperature fluctuations. The antenna can also be reconfigured to another shape beyond a temperature crossing event above T_2_. When the temperature is higher than T_2_, which indicates a further temperature crossing event, the blue section recovers its predefined straight shape, as shown in Figure 2c. This causes the sensor to form a parallel line configuration. An antenna with parallel arms does not radiate as the antiparallel currents in the two arms result in radiation that cancels out. Similarly, if the temperature drops below T_2_, the antenna still holds the parallel line shape. The historical temperature information can be known by sensing the radiation profile by measuring S_11_. Alternately, for applications in wireless communication, the temperature can be used as a stimulus to control the antenna configuration and its impedance (folded, dipole, etc.).

The shape variation caused by the temperature crossing event is detected by the reflection coefficient S_11_. When the temperature exceeds each of these thresholds, the reflection coefficient differs. To quantify the results, S_11_ of the different states of the SMA antenna is simulated. The total length of each multi-section SMA wire is 75 mm, which results in a magnitude peak at 950 MHz in the range of UHF RFID. One can also use an RF wireless reader to measure the change in received signal strength (RSSI) around its resonance frequency to identify the ShMoA configuration and thus the temperature crossing event information.

The simulated responses for the antenna with SMA dimensions in Figure 2 are shown in Figure 3. At a temperature below T_1_, the SMA sensor is deformed, as in Figure 2a, showing a U shape. Under this condition, the impedance of the antenna is 44 + 5 j Ω at 950 MHz. When the temperature is between T_1_ and T_2_, the antenna switches to a dipole antenna configuration (Figure 2b), and the impedance changes to 73 + 9 j Ω. Although a standard dipole is better at radiating geometrically, the improperly matched input impedance of the antenna will cause an overall weaker radiation [45,46]. As a result, the minimal magnitude of the return loss S_11_ around 950 MHz decreases from 23 dB to 16 dB. When further temperature crossing occurs and the temperature rises above T_2_, the SMA wires straighten and form a parallel line antenna (Figure 2c). At this point, the impedance of the antenna is 1–29 j Ω. The S_11_ flattens out at 0 dB over the frequency range of 0.7–1.2 GHz.

The radiation pattern of the ShMoA under the three states are also simulated at 940 MHz. Figure 4 shows the simulated 3D radiation plot with the corresponding antenna states and orientation. The radiation patterns of the three states are clearly distinguishable. Note that the scale of the color bars is different for each plot. The simulated radiation plane of the XZ and XY planes under all three states are plotted in Figure 5. For the XZ plane, the radiation is mostly symmetrical. For the XY plane, the cross-polarization is more than 20 dB less than the co-polarization when t < T_1_. As the temperature rises above T_1_ but below T_2_, the difference changes to less than 60 dB. As the temperature farther rises above T_2_, the cross-polarization and co-polarization become almost identical.

## 3. ShMoA Fabrication

The ShMoA designs, shown in Figure 2, are realized and experimentally validated. The SMA wires (Nitinol SMA, Nexmetal Corporation, Los Angeles, CA, USA) used in the design of the antenna have a diameter of 0.5 mm. The thresholds T_1_ and T_2_ of the two SMA wires are chosen to be 30 °C and 50 °C, respectively. To fabricate the multi-section return-loss tuned SMA antenna, the SMA wires must be electrically joined together. Soldering is preferred for its robust connection. However, the ordinary soldering method would fail due to an oxide layer forming on the surface of the SMA wire [47]. The heat during the soldering process greatly aerates the oxidation. Additionally, this oxidation layer prevents the solder from wetting the SMA wire, thus preventing the two SMA from joining together. To eliminate oxidation, sandpaper is used to polish the connecting end, and a brief treatment with an acid solution is applied to the polished part. This prevents oxidation and allows the two wire sections to be joined together. We show that a physically and electrically stable connection between the different sections of the SMA wire can be achieved using the aforementioned soldering approach. Afterward, the acid residue is removed, and the excess solder is trimmed to have a smooth connection. Testing is performed by attaching the multi-section SMA wires to a small printed circuit board with a coaxial connector. The performance of the fabricated ShMoA designs for temperature sensing will be discussed in the next section.

## 4. Experiments and Characterization

### 4.1. Temperature Response of Individual SMA Wires

The temperature response of the sensor is measured by visually recording the shape of the SMA wire while sweeping the temperature. To ensure the temperature of the environment temperature is steady, the fabricated SMA wire is placed in a water bath, which allows the temperature to be precisely controlled. A thermal coupler (RISEPRO) is placed nearby the SMA sensor to detect the temperature of the water bath. The fabricated multi-section SMA wire is initially cooled to 5 °C and shape-locked to a perpendicular shape shown in Figure 6a. The water bath is then placed on a hot plate and gradually heated from 5 °C to 60 °C, at a rate of approximately 2 °C/min. During the heating process, optical images of the SMA wires using conventional CMOS cameras are recorded as the wires change shape, and the angle of each section of the SMA wire is measured afterward using image processing. We define the angle of each section of the SMA as θ_1_ and θ_2,_ as shown in Figure 6a. The angles are converted to the percentage of the transition from the initial to the final angle value. In our experiment, the relationship between the percentile and the bending angle is as follows.
$$\mathrm{Shape}\;\mathrm{transition}\;\%=\frac{\theta -90{}^{\circ}}{180{}^{\circ}-90{}^{\circ}}\times 100\%$$

The connecting point of the SMA wire is fixed, while the two ends move freely as a function of temperature. The angle of each section is measured as the temperature increases.

Figure 6b shows the shape transition percentile of the two sections while the temperature increases from 20 °C to 60 °C. Initially, both sections are manually set to be 90°, which corresponds to a 0% shape transition. When the temperature is above their thresholds and the SMA wires become straight, the angles become 180°, which leads to a 100% angle transition. Section 1 has a threshold of around 30 °C, and Section 2 has a threshold of about 50 °C. In Figure 6b, the measurement temperature response of Section 1 is depicted as the blue dashed line. The shape of the SMA wire starts to change at 28 °C and finishes changing at 32 °C, resulting in a 4 °C transition zone. Similarly, the measured response of Section 2, shown as the solid red line, starts to drastically change shape at 48 °C and finishes changing at 53 °C. The transition zone is 5 °C.

### 4.2. Frequency Response of ShMoA vs. Temperature

The S_11_ of the actual fabricated SMA antenna at different temperatures is characterized by a Vector Network Analyzer (N5227A, Agilent, Santa Clara, CA, USA). The frequency responses of the SMA antennas are tested under three temperature ranges, which is below 30 °C, between 30 °C and 50 °C, and above 50 °C. The antenna is pre-heated to the corresponding temperature range and fixed in an anechoic chamber. Due to the memory effect, the antenna holds its shape, and the return loss can be measured. The figures of the SMA antennas under different states and the test results are shown in Figure 7 and Figure 8.

The measured frequency response of the ShMoA is shown in Figure 8. The ShMoA is in a U shape state when the temperature is below 30 °C, as shown in Figure 7a. Similar to the simulation results in Figure 3, the antenna in this state shows the largest reflection of 30 dB at 950 MHz. When the temperature is between 30 °C and 50 °C, the SMA sensor transforms into a dipole antenna, as shown in Figure 7b. As a result, the reflection coefficient S_11_ drops to 24 dB at 950 MHz. As the temperature rises above 50 °C, the SMA antenna turns into a parallel antenna, as shown in Figure 7c, and the S_11_ is less than 3 dB over the 0.7–1.2 GHz frequency range.

## 5. Discussion

The proposed approach to antenna reconfiguration with SMA achieves tuning of multiple antenna parameters through temperature-induced shape morphing. By conjoining various SMAs of different temperature thresholds together, the antenna can morph its shape and thus alter radiation pattern and strength according to different temperature events. Depending on the application, one can easily modify the design to accommodate the geometry for specific radiation patterns or frequency needs. The shape morphing mechanism is achieved through the thermal-mechanical memory effect of the SMA material from which the arms of the antenna are made. As a result, the ShMoA does not need any additional substrate or an actuator to achieve shape morphing. The antenna itself morphs into different configurations based on temperature. A dipole design of the ShMoA was demonstrated, which resembled an antenna used for temperature sensing in insects and flies. Such temperature sensing antennas can be deployed in mobile robots for thermal perception of its environment. While a simple design of folded dipole structure was shown, one can scale up the concept to add multiple SMA sections for a more comprehensive temperature sensing. Beyond sensing, such shape morphing structures can also support multiple wireless standards for communications with embedded frequency tuning and impedance matching. If deployed in a robot, the temperature can be used as an approach to program the antenna’s behavior automatically. Note that temperature change is only needed to initiate the shape change for reconfiguration; the design stays locked in a particular configuration as long as it remains in the said temperature range.

Since the shape of ShMoA depends on temperature, we demonstrated its use as a temperature sensor. Leveraging the SMA’s thermal hysteresis, the historical temperature information is recorded within the shape of the antenna itself, eliminating the need for an electronic memory or circuitry and thus forgoing the need for a battery. The reduced size, weight, area and power requirement makes it especially attractive for large scale deployment or disposable applications. Both simulation and experimental measurement show that the temperature event-induced shape-change can be monitored by the magnitude of the reflection coefficient S_11_ and the resonance frequency depending on the design. Once the cross event is detected, either the return loss or radiation pattern of the ShMoA changes as an indicator of the temperature crossing event. The prototype fabricated dual-section SMA arms of the SMA react to two different temperature thresholds. As indicated in Figure 6b, the first section will change from the manually set shape to a predefined shape due to the SMA’s thermal-mechanical memory effect. The transition-start temperature A_s_ is 28 °C, and the transition-finish temperature A_f_ is 32 °C. The 4 °C transition zone indicates a ±2 °C for the first section at 30 °C. The second section has a higher threshold. Its transition-start temperature is 48 °C, and the transition finishes at around 53 °C. The tolerance for the second section is ±3 °C. As a temperature sensor, ShMoA achieves only moderate accuracy. However, such a level of accuracy is quite adequate for temperature event monitoring for applications such as transportation of goods, where the goal is to provide some assurance of temperature levels, and the exact level of temperature threshold is not critical. It is quite possible to custom fabricate an SMA with smaller temperature transition error through specific techniques and material choices [48,49,50,51], which could be the focus of future work.

Moreover, the ShMoA shape tracks the peak temperature it experienced. Due to the hysteresis of the SMA, any temperature event will cause the shape of the antenna to change, and the shape is kept even after the temperature drops below the thresholds. In this paper, the 30 °C and 50 °C thresholds are chosen for demonstration. SMA wires of different temperature thresholds can be obtained by a specific material and dopant selection [50,51,52,53], and thus, one can program temperature thresholds. Note that the operating frequency is chosen to incorporate the widely used ultra-high frequency RFID band, which dictates the length of the antenna in the cm range. The TiNi alloy used in this work is a popular phase change material for MEMS applications [54]. Thin film sputtered TiNi has been extensively studied [55]. It is expected that smaller length of arms of the antenna will enable operation of ShMoA at higher frequencies with advantages of reduced dimensions of the overall antenna. For example, MEMS antenna in the dimension of around 10 mm will enable operation at 9.41 GHz [56], albeit with higher return loss.”

## 6. Conclusions

In this work, a ShMoA is presented that embeds temperature sensing, embedded computing for detection of threshold crossings, memory and radio-frequency-based readout, all in the single construct without the need for any battery or peripheral electronics. Utilizing the thermal-mechanical memory effect of the SMA, the antenna can morph its shape in accordance with the temperature events. The SMA wires serve as both the arms of the antenna arm and an actuator. No additional substrate is needed to manipulate antenna configuration. The antenna essentially combines the function of temperature sensing, embedded computing for detection of threshold crossings, and radio frequency readout, all in the single construct of a shape-morphing antenna (SmoA) without the need for any battery or peripheral electronics. One shape-morphing design was shown to alter the return loss of the antenna as an indicator of temperature event. With the hysteresis of the SMA, the shape change is maintained even after the clearance of the temperature event, in essence achieving sensing and recording without the need for a battery or peripheral electronics, except during readout. The proposed approach achieves a low-cost, easy-to-fabricate, and battery-free all-in-one shape-morphing solution for reconfigurable antennas. The design of the ShMoA can be scaled up to add multiple temperature-triggered shape transformations for multiple applications from temperature sensing in mobiles robots to programmable antennas capable of supporting multiple wireless communication standards.

## Figures and Tables

**Figure 1 micromachines-13-01673-f001:**
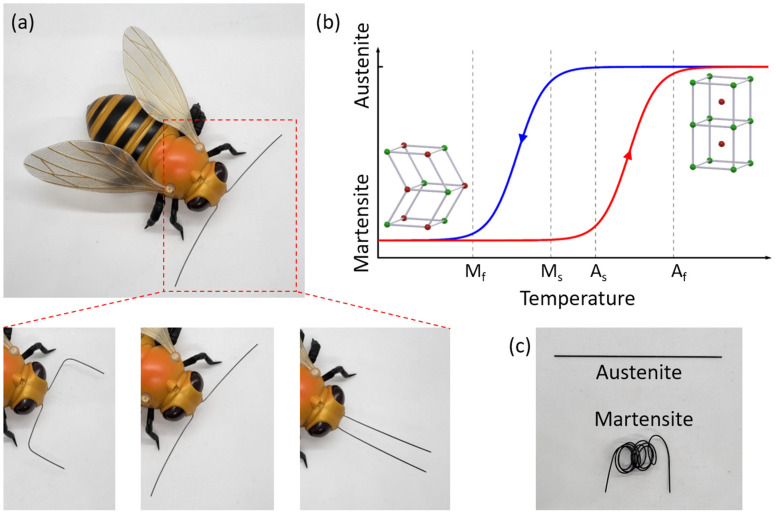
(**a**) ShMoA can be used for thermal perception in robots—changes shape based on temperature. (**b**) The hysteresis and critical temperature threshold of shape memory alloy is utilized for temperature sensing. The red curve is for heating process and the blue curve is for cooling process. (**c**) photo of SMA wires under martensite and austenite state at different temperatures.

**Figure 2 micromachines-13-01673-f002:**
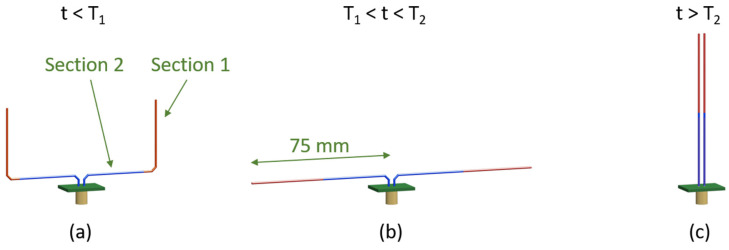
ShMoA under different temperature regions. Section 1 has a threshold temperature at T_1_; Section 2 has a threshold temperature at T_2_. (**a**) Temperature below both thresholds, (**b**) temperature crossing the first threshold, and (**c**) temperature crossing the second threshold.

**Figure 3 micromachines-13-01673-f003:**
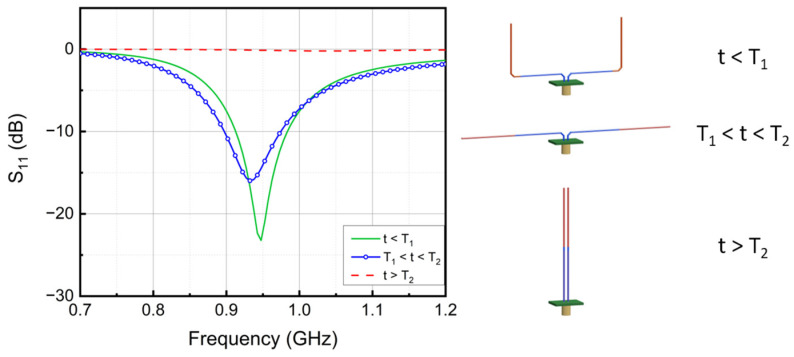
Simulated frequency response of the designed ShMoA and its corresponding shape at different temperatures.

**Figure 4 micromachines-13-01673-f004:**
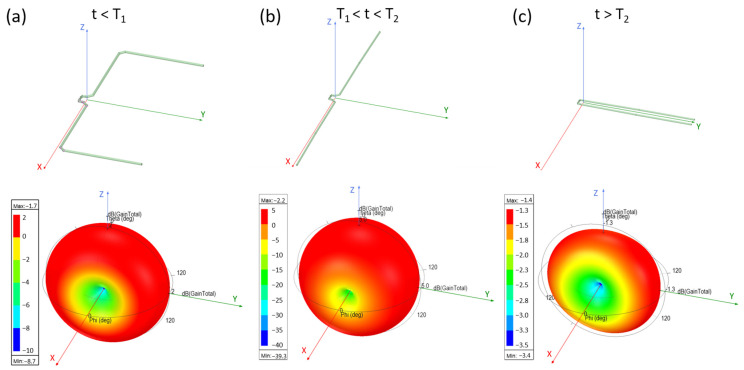
Simulated 3D radiation pattern of the ShMoA and corresponding antenna geometry at (**a**) t < T_1_, (**b**) T_1_ < t < T_2_, and (**c**) t > T_2_.

**Figure 5 micromachines-13-01673-f005:**
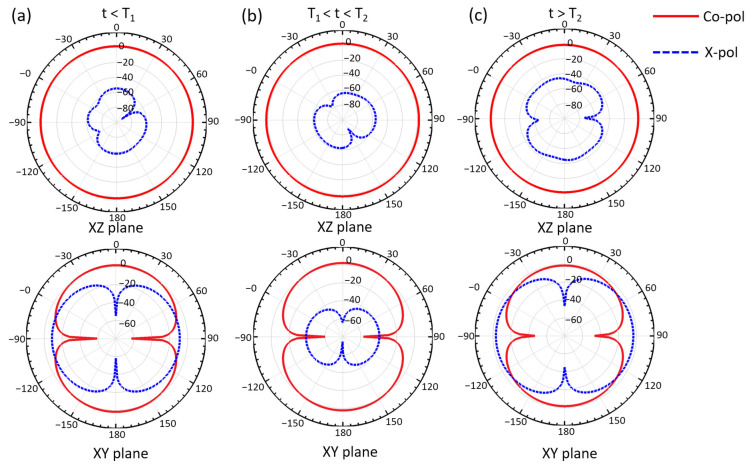
Simulated radiation pattern of the XY plane and XZ plane of the ShMoA at (**a**) t < T_1_, (**b**) T_1_ < t < T_2_, and (**c**) t > T_2_.

**Figure 6 micromachines-13-01673-f006:**
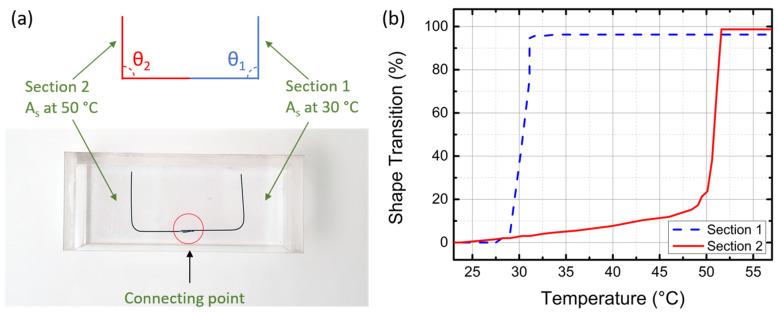
(**a**) Schematic and photo of multi-section SMA wire manually reset with an angle in both sections. (**b**) Temperature response of the SMA wire.

**Figure 7 micromachines-13-01673-f007:**
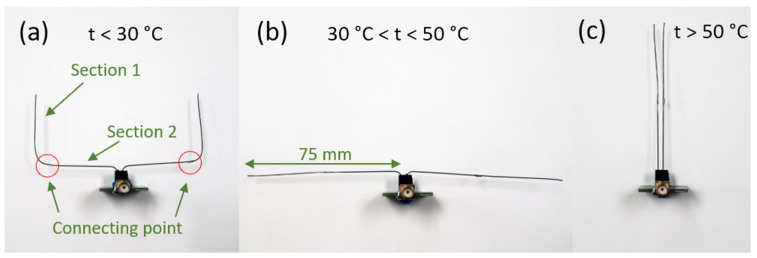
ShMoA under (**a**) no temperature crossing, t < 30 °C, (**b**) first temperature crossing event, 30 °C < t < 50 °C, and (**c**) the second temperature event, t > 50 °C.

**Figure 8 micromachines-13-01673-f008:**
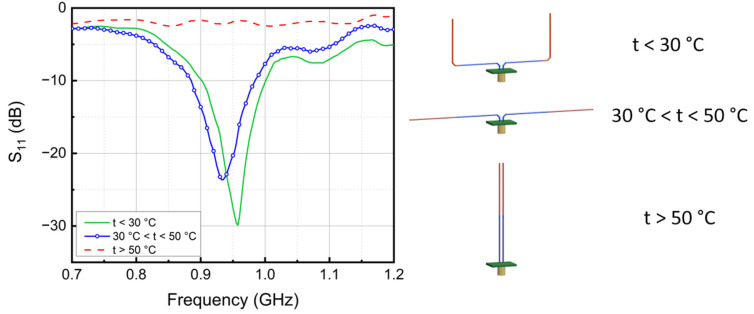
Measured frequency response of the ShMoA and its corresponding shape at different temperatures.

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
