# Peer review of "Temperature Sensing Shape Morphing Antenna (ShMoA)"

_micromachines, 2022, doi:10.3390/mi13101673_

Round 1

Reviewer 1 Report

General comments

The paper presents simulations and experimental validation of a shape-changing antenna based on SMA wires with different transformation temperatures. The paper is well written, very clear and provides a convincing demonstration of the proposed mechanism. It is relevant for the field.

The simulated and demonstrated device presented in the paper is on the mm to cm scale. As a publication in Micromachines, the authors must address the size-scaling of this mechanism. Specifically, can it be feasibly down-scaled to MEMS-based devices, and if yes how? This point must be addressed.

Additional revisions to be addressed prior to publication:

1.     Can the authors evaluate the maximal strain recovered in the SMA elements as a result of the shape change from a bent to a straight wire? Can they comment on the expected fatigue life of such a device? How do the authors track and quantify the angle theta? Is this done using fast optical imaging and image analysis? What is the approximate heating rate of the bath?

2.     Authors should elaborate on the frequency response measurements. How was the antenna heated and its temperature controlled in this case?

Minor revisions:

1.     Page 3, Figure 1(b): the vertical axis in such a representation typically represents the strain associated with the phase transformation. This should be noted. In addition, the word “Austenite” on the vertical axis is cut.

2.     The sentence in the paragraph before figure 3: “Fig. 3 shows the simulated response of the two configurations” is redundant and appears again immediately after Fig 3.

Reviewer 2 Report

This paper shows a SMA-based antenna. The work is interesting, but the paper is unsuitable for publication. The main problem exists in its working approach. The SMA remains its shape change after the high temperature event, which may be a great obstacle in its application. A mechanism should be found to automatically recover its original state, or the device will become a manually operating/single-use device. A practical application can be very helpful to verify the real value of this paper.

Then, the authors have published several papers on the temperature-depended antenna, it is necessary to distinguish the newly submitted one from published ones. Though additional temperature threshold is added, I am still not very convinced about the novelty of this paper after reading all these published papers.

Lastly, the paper writing should also be improved. The full information of authors should be provided according to the editorial style. It is not necessary to continually write the full spelling of ShMoA after defining it in abstract.  

Round 2

Reviewer 2 Report

Thank you for the response.

 I’m still interested in the practical application of ShMoA. Why not simply using a temperature sensor for the measurement? As a temperature-depend antenna, where and how we use it?  Necessary explanations should be added in the discussion section.  

Then, as title said the device is a “Programmable” device. However, the device only has three states and cannot recover to the original shape by itself. It may be not very exact to use “Programmable” in the title.
